# A Case of Unilateral Hyperpigmentation

Yujin Han [1], Se-Hoon Lee [1], Minah Cho [1], Sang-Hyun Cho [1], Jeong-Deuk Lee [1], Yu-Ri Woo [1] and Hei-Sung Kim [1,2,*]

1   Department of Dermatology, Incheon St. Mary's Hospital, The Catholic University of Korea, Seoul 06591, Korea; thyme3700@gmail.com (Y.H.); leesehoon92@gmail.com (S.-H.L.); macho_maria@naver.com (M.C.); drchos@yahoo.co.kr (S.-H.C.); leejd@catholic.ac.kr (J.-D.L.); w1206@naver.com (Y.-R.W.)

2   Department of Biomedicine & Health Sciences, The Catholic University of Korea, 222 Banpo-dearo, Seocho-gu, Seoul 06591, Korea

*   Correspondence: hazelkimhoho@gmail.com; Tel.: +82-032-280-5104

**Abstract:** Phytophotodermatitis is a cutaneous phototoxic reaction resulting from contact with plant compounds such as furocoumarin in citrus fruits, followed by exposure to ultraviolet light. Erythema and vesicles appear on the contact area, followed by hyperpigmented lesions. Hyperpigmentation may exist for weeks to months before fading but can remain up to several years. Diagnosis is often challenging due to the variety of clinical presentations, and it is not always easy to identify trigger exposures. A detailed history is key to diagnosis. We herein report a case of lime-induced phytophotodermatitis which was initially mistaken for unilateral lentiginosis. The patient underwent Q-switched Nd:YAG laser and intense pulsed light (IPL) treatment with immediate improvement.

**Keywords:** furocoumarin; hyperpigmentation; intense pulsed light; lime; phytophotodermatitis; Q-switched Nd:YAG laser; unilateral

## 1. Introduction

Phytophotodermatitis is a non-immunologic cutaneous reaction caused by exposure to photosensitizing compounds in plants and subsequent exposure to sunlight [1]. Furocoumarin is a photosensitizing substance produced by certain plants including lime, celery, fig and parsnip [2]. Clinical findings evolve from the initial painful erythema and edema to blisters and late hyperpigmentation. Diagnosis is challenging and is often misdiagnosed as other skin conditions including allergic contact dermatitis and burns (early) and pigmentary disorders (late), especially when a patient fails to mention the relevant background. Treatment is symptomatic, the rash being resolved in weeks, though hyperpigmentation can persist for months. Herein, we present a case of lime-induced phytophotodermatitis which was initially mistaken for unilateral lentiginosis and showed drastic improvement after a single session of Q-switched Nd:YAG laser and intense pulsed light treatment.

## 2. Case

A 26-year-old woman presented with asymptomatic, diffuse, brown macules and patches on the right hand and arms for weeks (Figure 1). The patient did not have any past medical or dermatologic problems. On histopathologic examination, mild elongation of the rete ridges and increased pigmentation of the basal keratinocytes with few scattered dermal melanophages were shown (Figure 2). Based on the brief clinical exam and pathology, the patient was presumed to have unilateral lentiginosis and underwent a single session of intense pulsed light (IPL) and Q-switched neodymium-doped yttrium aluminum garnet (Nd:YAG) laser treatment. The patient was advised to use sunscreen to minimize laser-related post-inflammatory hyperpigmentation (PIH).

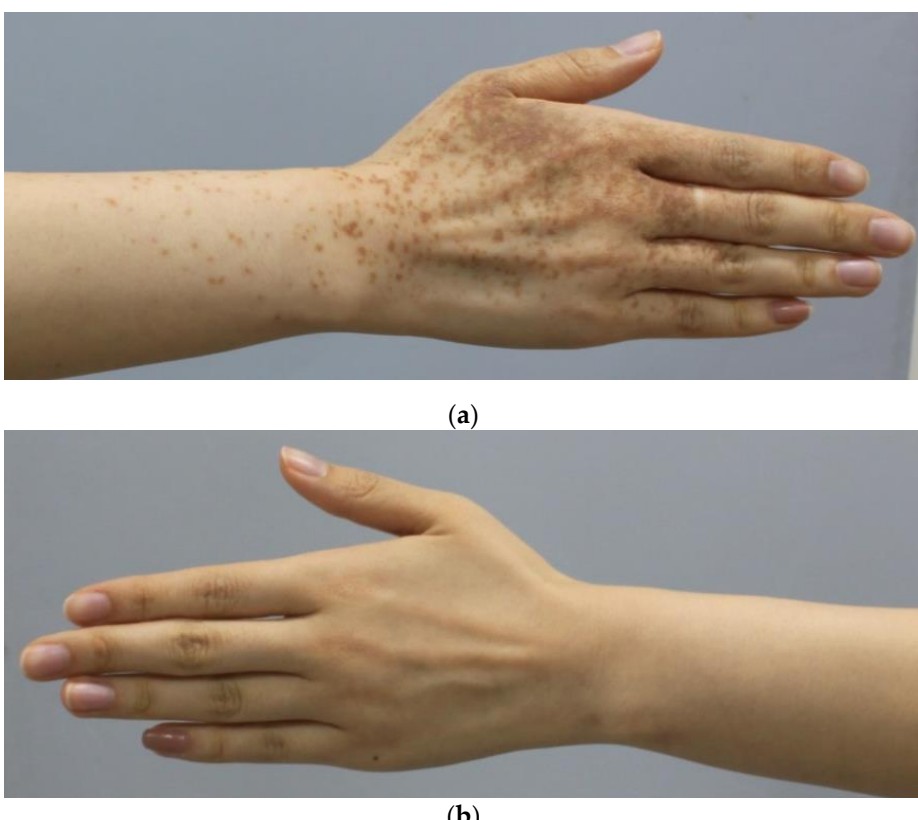

(**a**)

(**b**)

**Figure 1.** (**a**) Right arm (lesion): There are diffuse brown macules and patches, sparing the area under the ring on the third finger. (**b**) Left arm (control).

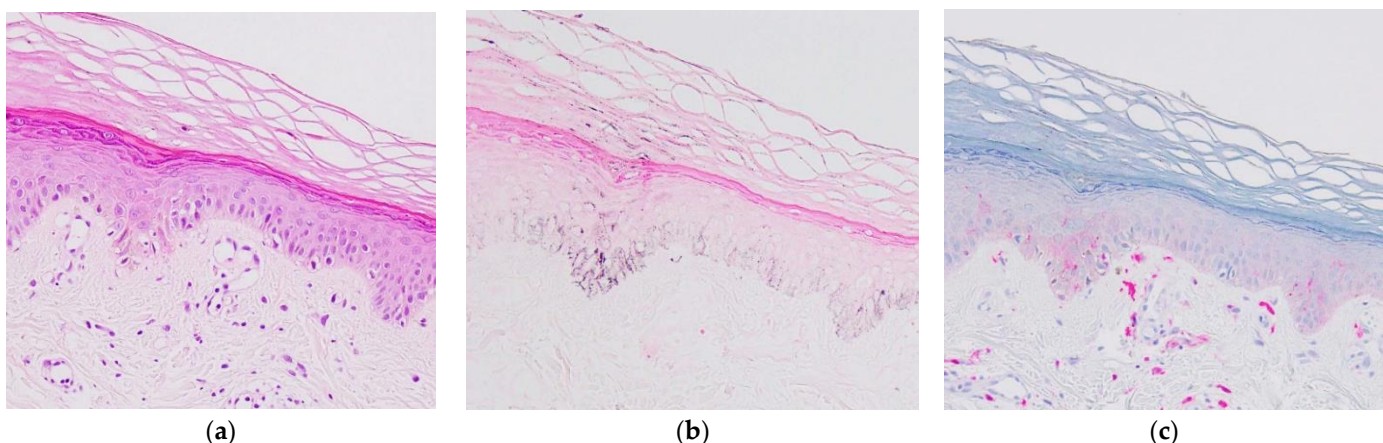

(**a**) (**b**) (**c**)

**Figure 2.** (**a**) Histopathological examination of the lesion reveals mild rete ridge elongation, increased pigmentation of basal keratinocytes and scattered dermal melanophages (H&E stain, ×40). (**b**) Increase in basal melanin (Fontana-Masson) and (**c**) Dermal melanophages (S-100) are seen.

After two months, the hyperpigmented lesions showed significant improvement, which was surprising considering the nature of lentigines which normally require multiple treatment (Figure 3). On further history taking, the patient was said to have made a trip to Vietnam where she squeezed and touched lime exclusively with her right hand. The patient revealed that the area first showed erythema and a few vesicles which later-on turned into pigmented spots. From the history and prompt response to treatment we made a final diagnosis of lime-induced phytophotodermatitis, which was initially overlooked.

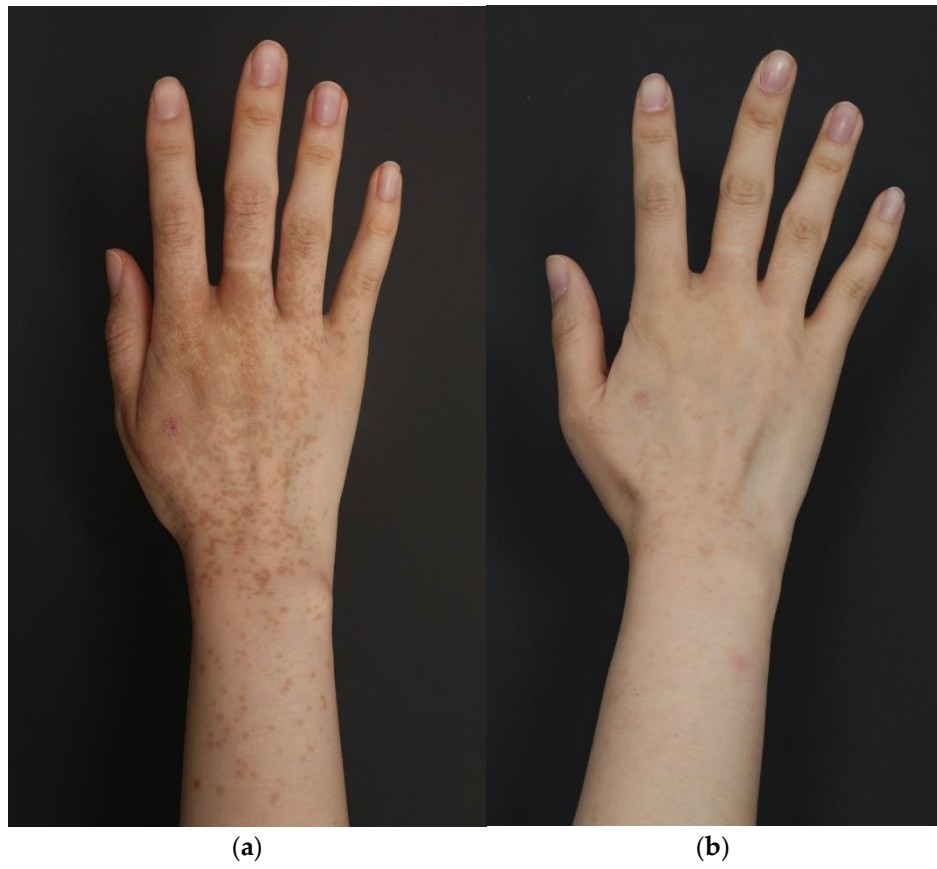

(**a**) (**b**)

**Figure 3.** (**a**) Before laser treatment and (**b**) After a single session of Q-switched Nd:YAG and IPL, at 2 months follow-up.

## 3. Discussion

Phytophotodermatitis is a phototoxic reaction that occurs after exposure to plant-derived photosensitizing compounds followed by ultraviolet A (UVA 320–380 nm) radiation [1]. Various plants including citrus fruit, celery, wild parsnip, carrot, and fig contain phototoxic agents that precipitate phototoxic reactions [2]. Furocoumarin is such a compound and normally protects plants against the attack of fungal pathogens [3]. When furocoumarin is ingested or applied to the skin with the skin being further exposed to UVA, cellular damage is triggered, either directly through a photosensitizer-target interaction or indirectly through the generation of reactive oxygen species (ROS) [4].

After the first description of phytophotodermatitis in 1942 by Klaber [5], there have been a number of case reports on lime-induced phyophotodermatitis (Table 1). The epidemiology is not well known, but most cases have been reported in cultures that widely consume limes (i.e., Mojito).

Phytophotodermatitis usually begin as erythema with subsequent blistering of the contact area, 12–36 h after furocoumarin exposure to UVA. Due to melanocyte hypertrophy, increased melanosome density and melanocyte dendricity, and dermal melanophages, hyperpigmented lesions may appear [6].

A detailed patient history is important in making a diagnosis, but patients sometimes do not disclose information because they think it is not relevant. Thus, it is important for the clinician to be aware of phytophotodermatitis because it is often misdiagnosed as allergic contact dermatitis, cellulitis, herpes simplex virus infection and even child abuse (early) and pigmentary disorders (late) [7].

There is limited evidence on the efficacy of treatment. Treatment is largely symptomatic, where a cold compress and topical steroids are sufficient for mild symptoms, whereas systemic steroids are indicated in severe cases. Hyperpigmentation may exist

for weeks to months before fading but can persist up to several years if not properly managed [8].

Treatment options for the hyperpigmentation in phytophotodermatitis are similar to that of PIH from other causes. Topical agents such as hydroquinone and retinoids can be applied. Lasers that target melanin such as Q-switched Nd:YAG laser, as well as IPL, can be used successfully in persistent cases [9,10]. Although not routinely recommended, Q-switched Nd:YAG laser can be an option for those who want to improve their hyperpigmentation in a short period of time. In our case, the lesions disappeared shortly after a single session of Q-switched Nd:YAG and IPL.

**Table 1.** Review of lime-induced phytophotodermatitis in the literature.

| No. | Publication | Age (yr)/Sex | Cutaneous Findings | Location | Contact History |
|---|---|---|---|---|---|
| 1 | Coffman et al. [11] 1985 | 4/F | Hyperpigmentation | Chest, upper back, and wrist | Squeezing lime |
| 2 | Coffman et al. [11] 1985 | 1/M | Hyperpigmentation | Chest and back | Squeezing lime |
| 3 | Wagner et al. [12] 2002 | 6/M | Erythema and bullae | Hand | Squeezing lime |
| 4 | Pomeranz et al. [1] 2007 | 23/F | Erythema and bullae | Back, arm, and hand | Preparing mojitos |
| 3 | Hankinson et al. [13] 2014 | 24/F | Erythema and bullae | Hand | Squeezing lime while cooking |
| 4 | Mioduszewski and Beecker [14] 2015 | 26/F | Erythema and bullae | Hand | Squeezing lime while making sangria |
| 5 | Galvañ-Pérez et al. [15] 2016 | 23/F | Hyperpigmentation | Hand | Preparing mojitos |
| 6 | Galvañ-Pérez et al. [15] 2016 | 25/F | Hyperpigmentation | Hand | Preparing mojitos |
| 7 | Galvañ-Pérez et al. [15] 2016 | 31/F | Hyperpigmentation | Hand | Preparing mojitos |
| 8 | Galvañ-Pérez et al. [15] 2016 | 19/F | Hyperpigmentation | Hand | Preparing mojitos |
| 9 | Galvañ-Pérez et al. [15] 2016 | 41/M | Hyperpigmentation | Hand | Preparing mojitos |
| 10 | Galvañ-Pérez et al. [15] 2016 | 40/M | Hyperpigmentation | Hand | Preparing mojitos |
| 11 | Galvañ-Pérez et al. [15] 2016 | 21/F | Hyperpigmentation | Hand | Preparing mojitos |
| 12 | Galvañ-Pérez et al. [15] 2016 | 14/F | Hyperpigmentation | Hand | Preparing mojitos |
| 13 | Galvañ-Pérez et al. [15] 2016 | 16/F | Hyperpigmentation | Hand | Preparing mojitos |
| 14 | Snaidr and Lowe [16] 2017 | 36/F | Erythema and bullae | Hand | Lime juice |
| 15 | Safran et al. [17] 2017 | 17/F | Erythema and bullae | Hand | Squeezing lime while cooking |
| 16 | Fitzpatrick et al. [18] 2018 | 31/M | Hyperpigmented macules | Hand | Squeezing lime while mixing margaritas |
| 17 | Abugroun A et al. [2] 2019 | 26/M | Erythema and bullae | Hand | Squeezing lime |
| 18 | Wang and Ma [19] 2021 | 6/F | Hyperpigmented macules | Shoulder | Squeezing lime during the hot spring |
| 19 | Wang and Ma [19] 2021 | -/M (No.18's father) | Hyperpigmented macules | Upper back | Squeezing lime during the hot spring |
| 20 | Hamid et al. [20] 2021 | 25/M | Hyperpigmented macules | Hand | Drinking beer with lime |

If contact to the plants was recognized, the skin should be washed with water before exposure to sunlight. Additionally, as a preventive measure, we recommend vigorous photoprotection (i.e., sunscreen, physical sunshields) after having contact with plants containing furocoumarin. The importance of sunscreen is also emphasized after laser treatment to minimize the risk of PIH.

**Author Contributions:** Conceptualization, H.-S.K.; methodology, Y.H.; validation, S.-H.C., J.-D.L. and Y.-R.W.; formal analysis, Y.H.; investigation, Y.H., S.-H.L.; resources, M.C.; data curation, Y.H.; writing—original draft preparation, Y.H.; writing—review and editing, Y.-R.W., H.-S.K.; visualization, Y.H.; supervision, H.-S.K.; project administration, M.C. All authors have read and agreed to the published version of the manuscript.

**Funding:** This research received no external funding.

**Institutional Review Board Statement:** The study was conducted according to the guidelines of the Declaration of Helsinki, and approved by the Institutional Review Board of Incheon St. Mary's hospital, Incheon, South Korea (OC21ZISI0047, 21 May 2021).

**Informed Consent Statement:** Informed consent was obtained from all subjects involved in the study.

**Data Availability Statement:** Not applicable.

**Conflicts of Interest:** The authors declare no conflict of interest.

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
