# Peer review of "A Case of Unilateral Hyperpigmentation"

_dermato, doi:10.3390/dermato1010004_

Round 1

Reviewer 1 Report

Line 8. …is “a” cutaneous..

Line 25. Instead of saying the relevant history, it is better to say the relevant background.

Line 25. Being the rash resolved

Figure 1. You could omit the figure 1C. The figure foot should include 1) left arm and 2) right arm. You could enlarge and trim the extent of the lesions.

Line 29. You could include more information regarding patient history

Line 35. After two months…

Are there some histopathological features that could help clinicians to differentiate unilateral lentiginosis from phytophotodermatitis? Why were not the patients firstly asked for background instead of taking a biopsy?

It would be interesting to make a table that included the lime-induced photodermatitis reported in the literature

Author Response

Line 8. …is “a” cutaneous..

Authors’ response> Thank you for the comment. It has been modified as requested by the reviewer.

Line 25. Instead of saying the relevant history, it is better to say the relevant background.

Authors’ response> It has been corrected.

Line 25. Being the rash resolved

Authors’ response>

We have modified the sentence to “Treatment is symptomatic, the rash being resolved in weeks…”

Figure 1. You could omit the figure 1C. The figure foot should include 1) left arm and 2) right arm. You could enlarge and trim the extent of the lesions.

Authors’ response> We have modified the figure as requested.

Line 29. You could include more information regarding patient history.

Authors’ response> We have added that the patient did not have any past medical or dermatological conditions.

Line 35. After two months…

Authors’ response> We have corrected it.

Are there some histopathological features that could help clinicians to differentiate unilateral lentiginosis from phytophotodermatitis? Why were not the patients firstly asked for background instead of taking a biopsy?

Authors’ response>

In unilateral lentiginosis, slight to moderate elongation of the epidermis ridges associated with an increased number of basal melanocytes are observed. On the other hand, the chronic lesion of phytophotodermatitis shows patchy epidermal melanosis with dermal melanophages.

In Far East Asia, phytophotodermatitis is not so common and we overlooked the possibility of the disorder presuming that it was unilateral lentiginosis. We are reporting this case in hope for others not to make the same mistake in future performing unnecessary biopsies.

It would be interesting to make a table that included the lime-induced photodermatitis reported in the lit erature.

Authors’ response> Thank you for the comment. We have summarized the reported literature in Table 1.

Reviewer 2 Report

Dear Dr. Han, 

I have read your case of unilateral hyper pigmentation with great interest. 

Please find below some minor comments: 

  • Did patient also present hyper pigmentation palmar?
  • Does protection of natural UV light has a positive effect on lime induced hyperpigmentation? If so please give a short comment to that. 
  • Please give any comment / additional information if this woman in your case took any addition systemic or topical medication (which might have an impact on the development of photodermatitis. 

Author Response

I have read your case of unilateral hyper pigmentation with great interest. 

Please find below some minor comments: 

Did patient also present hyper pigmentation palmar?

Authors’ response>

Thank you for the comment. The palm was not involved. After squeezing the lime, the patient reportedly rubbed it onto the right-hand dorsum which was subsequently exposed to strong sunlight.

Does protection of natural UV light has a positive effect on lime induced hyperpigmentation? If so please give a short comment to that. 

Authors’ response>

Since UV is required for phytophotodermatitis to happen, we believe that the protection of natural UV light would have a preventive effect on lime induced hyperpigmentation. We have made a short comment on this in discussion.

Please give any comment / additional information if this woman in your case took any addition systemic or topical medication (which might have an impact on the development of photodermatitis. 

Authors’ response>

Thank you for the comment. Our patient denied of taking any oral or topical medication.

Reviewer 3 Report

The paper entitled “ A case of Unilateral Hyperpigmentation” reports a case of atypical photo-inducted unilateral hyperpigmentation, successfully treated with Q-switched Nd:YAG 1064 nm laser.

The case is well documented and may be of interest. However, some points need to be resolved before publication.

The treatment performed, which resulted in an evident clinical benefit, should be mentioned both in the abstract and in the final part of the introduction.

Similarly, diagnostic challenges should be mentioned in the abstract.

It was not specified whether the patient also received indications to the need for photoprotection. The authors should also discuss how much subsequent photoprotection can affect the response to the treatment.

In the discussion, it could be interesting to report the response rates to the various treatments.

Author Response

The paper entitled “A case of Unilateral Hyperpigmentation” reports a case of atypical photo-inducted unilateral hyperpigmentation, successfully treated with Q-switched Nd:YAG 1064 nm laser.

The case is well documented and may be of interest. However, some points need to be resolved before publication.

The treatment performed, which resulted in an evident clinical benefit, should be mentioned both in the abstract and in the final part of the introduction.

Authors’ response>

Thank you for the comment. We have mentioned the laser treatment we performed on our patient in the abstract and introduction as requested.  

Similarly, diagnostic challenges should be mentioned in the abstract.

Authors’ response>

Thank you for the comment. Diagnosis is often challenging due to the variety of clinical presentations and not always easily identified trigger exposures. We have mentioned this in the abstract.

It was not specified whether the patient also received indications to the need for photoprotection. The authors should also discuss how much subsequent photoprotection can affect the response to the treatment.

Authors’ response>

Thank you for the comment. The patient was asked to apply sunscreen continuously after laser treatment. Photoprotection minimizes the risk of post-inflammatory pigmentation after laser treatment which we have mentioned in discussion.

In the discussion, it could be interesting to report the response rates to the various treatments.

Authors’ response>

Thank you for the comment. The post-inflammatory pigmentation which follows phytophotodermatitis doesn’t routinely require treatment, but dermatologists can provide an option of topicals (i.e. vitamin A cream, hydroquinone) or laser therapy (i.e. Q-switched lasers to IPL) for those who want fast improvement. Unfortunately, the response rate to treatment has not been properly evaluated in prior reports.

Round 2

Reviewer 3 Report

Changes requested by the reviewer have been made.  English spell check is required.